# Federated Learning for Diabetic Retinopathy Detection Using Vision Transformers

**Mohamed Chetoui** and **Moulay A. Akhloufi** *

Perception, Robotics and Intelligent Machines Research Group (PRIME), Department of Computer Science, Université de Moncton, Moncton, NB E1A 3E9, Canada; emc7409@umoncton.ca
* Correspondence: moulay.akhloufi@umoncton.ca

**Abstract:** A common consequence of diabetes mellitus called diabetic retinopathy (DR) results in lesions on the retina that impair vision. It can cause blindness if not detected in time. Unfortunately, DR cannot be reversed, and treatment simply keeps eyesight intact. The risk of vision loss can be considerably decreased with early detection and treatment of DR. Ophtalmologists must manually diagnose DR retinal fundus images, which takes time, effort, and is cost-consuming. It is also more prone to error than computer-aided diagnosis methods. Deep learning has recently become one of the methods used most frequently to improve performance in a variety of fields, including medical image analysis and classification. In this paper, we develop a federated learning approach to detect diabetic retinopathy using four distributed institutions in order to build a robust model. Our federated learning approach is based on Vision Transformer architecture to classify DR and Normal cases. Several performance measures were used such as accuracy, area under the curve (AUC), sensitivity and specificity. The results show an improvement of up to 3% in terms of accuracy with the proposed federated learning technique. The technique also resolving crucial issues like data security, data access rights, and data protection.

**Keywords:** federated learning; Vision Transformers; diabetic retinopathy; medical imaging

## 1. Introduction

In recent years, deep learning (DL) has been widely used to streamline processes in the medical industry. The expertise and popularity of research and development in DL have increased dramatically in areas like disease screening systems, automated diagnosis, prognosis or treatment prediction [1–3], and smart health care [4], which have the potential to significantly enhance clinical workflow [5,6]. With the aid of image-based data such as retinal images, optical coherence tomography (OCT) images, and OCT angiography (OCTA) images, DL algorithms have been developed in ophthalmology to detect and classify a variety of ocular diseases, including diabetic retinal diseases [7,8], age-related macular degeneration [9,10], retinopathy of prematurity [11], and glaucomatous optic neuropathy [12–14].

The development of DL algorithms also demonstrated their capacity for using retinal images to diagnose and predict serious illnesses such diabetes [15], chronic kidney disease [16], cardiovascular events [17], and Alzheimer's disease [18]. Additionally, DL-based ocular image processing can be combined with telemedicine to detect and track eye disease in patients receiving primary care and treatment in community clinics [19]. Deep learning requires a large and diverse training dataset collection to increase robustness and generalizability. In order to create DL algorithms that are workable in many real-world scenarios, multicenter research is becoming more and more crucial [20,21]. The "centralized learning" paradigm is currently the most popular model for such collaborative multicenter initiatives, in which data from several sites are sent to and gathered in one location in accordance with inter-site agreements. Big data gathering and resource sharing could, however, cause practical complications, and it frequently takes time to find solutions to ethical

and privacy-related problems. Even anonymous raw images used in medical imaging can contain sensitive patient data. Since age [22], sex [23], cardiovascular risk factors [17], or mortality risk [24] might be predicted from fundus images or OCT scans, retinal images, for example, are as distinctive as fingerprints [25] and very sensitive. De-identified magnetic resonance imaging (MRI) images can be used to rebuild human faces [26]. Therefore, the "distributed learning" paradigm [27] has been established to divide data among multiple institutions rather than combining it into a single pool in order to preserve data privacy and eliminate the potential danger of raw data leakage in the conventional paradigm (i.e., centralized learning). Federated learning (FL) [28–30], a recent development in distributed learning, enables many medical institutions to cooperatively train AI models without data exchange. It greatly facilitates AI research and development in the healthcare industry, where access to vast amounts of data and various centres are often required due to the high value of the data. For model training and testing, the traditional DL approach calls for combining all accessible data from several institutions into a single source. On the other hand, FL uses a distributed learning paradigm in which numerous participants train a model locally on their own data and then send model changes to a central server to be combined into a consensus model [31]. It eliminates the need to centralize all acquired data or provide collaborators immediate access to sensitive data. No data are transferred or directly accessed between institutions; instead, each one retains its data locally.

The FL paradigm for model training is based on three main steps: (i) the global model is initially initialized by the central server and then distributed to each contributing institution; (ii) each institution trains the mode using its local data and then sends the local model back to the central server; and (iii) the central server aggregates all local models to update a new global model and redistributes it to all collaborators. Until the global model performs consistently, these processes are repeated back and forth. In both standard DL and FL, the model training process is the same. However, the sole distinction between the DL and FL training paradigms is that the DL requires that a central institution train the model on all data, whereas FL permits local training by each institution. This distributed training strategy has an enormous amount of potential to protect data privacy among many institutions and avoid the possible risk of data leakage from data centralization because only the model characteristics (such as model parameters or gradients) are to be sent out from institutions. To increase the model's resilience and generalizability, it can be trained and validated using numerous datasets. FL therefore offers enormous data privacy advantages over traditional centralized learning systems, particularly in AI research and the healthcare industry.

Numerous applications for medical image analysis currently employ FL. This distributed learning approach can build a robust model using large and varied medical imaging datasets obtained from numerous institutions while maintaining patient privacy and data ownership. Its potential for detecting various retinal illnesses using ocular images like OCT and retinal fundus imaging has already been demonstrated in ophthalmology.

For the classification of referable diabetic retinopathy (RDR) utilizing OCT and OCTA from two separate institutions, Yu et al. [32] used the FL framework. The FL model's performance was compared to the models trained using data from the same institution and from different institutions. The performance was superior to that of those trained on data from other institutes and on par with that trained on local data. The study showed thar FL may be used to classify DR and make it easier for diverse institutions to work together in the real world. The study additionally examined the FL method for applying microvasculature segmentation to various datasets in a simulated setting. The authors created a reliable FL framework for segmenting the microvasculature in OCTA images. Four distinct OCT devices were used to collect the image datasets. Their FL models achieved an accuracy (ACC) score of 0.762–0.880, an F1 score of 0.677–0.909 and an AUC of 0.910–0.979; the internal models achieved performances of 0.809–0.908, 0.778–0.921, and 0.884–0.978, respectively.

The disease known as retinopathy of prematurity (ROP), which is one of the main causes of young blindness globally, is distinguished by the development of abnormal fibrovascular retinal structures in premature infants. A deep learning model for ROP was developed using the FL technique, which was investigated by Hanif et al. [33] and Lu et al. [34]. Utilizing 5245 ROP retinal images from 1686 eyes of 867 preterm children in neonatal intensive care at seven hospital centres in the United States, Lu et al. [34] trained and validated DL models. Three image-based ROP graders assigned the images as clinical diagnoses of plus illness (plus, pre plus, or no plus), as well as a reference standard diagnosis (RSD). With an AUC ranging from 0.93 to 0.96, the models trained using the FL technique performed on par with those trained using the central learning strategy in the majority of DL model comparisons. Additionally, utilizing only a single institution's worth of data, the FL model outperformed the locally trained model in terms of AUC score in four of seven sites. The FL model maintained its accuracy and consistency across heterogeneous clinical data sets from various institutions, which differed in sample sizes, disease prevalence, and patient demographics.

The potential of FL to reconcile the disparity in clinical diagnosis of ROP severity between institutions was shown in the second trial by Hanif et al. [33]. An FL model was created based on the ROP vascular severity score (VSS) rather than the consensus RSD. The amount of VSS in the study's eyes with no additional illness differed significantly from those with the condition. VSS might be arbitrary, with wide variations across experts in clinical settings that could have an impact on epidemiology or clinical research [35]. The researchers discovered that there were substantial institutional differences in the number of patients with preplus illness ($p < 0.001$). They discovered disparities in the institutional VSS and the level of vascular severity classified as no plus ($p < 0.001$) among institutions using the DL-derived VSS trained on the data from all institutions using FL. The institutional VSS and mean gestational age showed a significant, inverse connection ($p = 0.049$, adjusted R2 = 0.49).

Lo et al. [36] presented federated learning for microvasculature segmentation and diabetic retinopathy classification using OCT images. The DR classification was divided into the non-RDR and RDR classes. The simulation used four clients to demonstrate the federated learning configuration for microvasculature segmentation and to compare it to other collaborative training techniques. For RDR classification, federated learning was used across several institutions, and it was compared to models that were trained and tested using data from the same institution (internal models) and from different institutions (external models), respectively. Federated learning provided results that were comparable to internal models for both applications. The federated learning model specifically performed similarly for microvasculature segmentation (mean DSC across all test sets, 0.793) to models trained on a fully centralized dataset (mean DSC, 0.807). The internal models attained a mean AUC of 0.956 and 0.973 for RDR classification, while federated learning attained a mean AUC of 0.954 and 0.960. The other derived assessment indicators show similar results.

Nasajabour et al. [37] investigated three models using standard transfer learning, Federated Averaging (FedAVG), and Federated Proximal (FedProx) frameworks. The authors demonstrated that the three models, including standard, FedAVG, and FedProx, are able to detect DR and non-DR images with an ACC of 0.92, 0.90, and 0.85, respectively.

Mohan et al. [38] proposed a DR severity grading technique based on FL. They combined the Federated Averaging technique and the median of the categorical cross-entropy loss. The authors proposed a central server to extract the features from the fundus images in order to identify the DR signs. In their study, they considered five clients holding different preprocessed fundus images obtained from public dataset such as MESSIDOR-2 [39], IDRID [40], Kaggle [41], and a local dataset collected from the Silchar Medical College and Hospital. Their proposed approach obtained an ACC score of 0.98, a specificity of 0.99, a precision of 0.97 and an F1 score of 0.97.

The FL model offers a generalizable method for evaluating clinical diagnostic paradigms and severity of illness for epidemiologic review without disclosing patient information, according to the study's findings.

In this study, we provide a novel approach for FL based on Vision Transformers (FLViT) for DR classification (DR and Normal). The approach allows clients the sharing of their ViT models with the central server in order to build a robust model. Our study solves the problem of confidentiality and data security. Finally, we compare the performance of the models trained with FLViT and local trained models (Non-FLViT). Our contribution includes the development of Transformer vision models that are trained locally and shared with the server; the server aggregates the ViT models according to the federated learning process for the task of DR classification using the fundus images. Our objective is also to investigate the performance of this technique, as none of the studies to date have ever integrated the Transformer vision models into their FL framework to detect diabetic retinopathy. In our study, we use four publicly available datasets: APTOS [42], MESSIDOR-1 and MESSIDOR-2 (merged into a single dataset), IDRiD and Eyepacs. Each dataset is assumed to be assigned to an institution. Figure 1 offers an overview of our methodology.

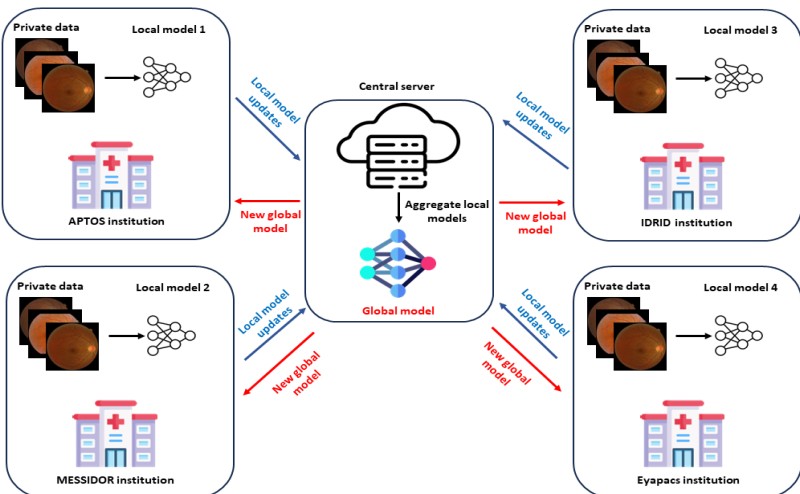

**Figure 1.** Proposed federated learning architecture for DR detection.

## 2. Vision Transformer

Vision Transformer (ViT) is a deep neural network based on an attention mechanism that utilizes a remarkably wide receptive field. The ability to attain state-of-the-art (SOTA) performance in NLP and the ability to represent long-range dependency inside an image have inspired the vision community to investigate exploring its use for vision problems [43]. The ViT was one of the successful attempts to apply transformers directly on images, and it was compared to SOTA convolutional neural networks in image classification tests [44]. In addition to its great performance, ViT's straightforward modular design offers vast applicability in a range of tasks with little modification. Chen et al. [45] presented an image processing transformer, one of the successful multi-task models for various computer vision tasks, by segmenting ViT into a shared body and task-specific heads and tails. The encoder–decoder technique was applied. ViT's SOTA performance was recently demonstrated when it was used to identify and forecast the severity of diabetic retinopathy [46–48].

Transformers come in a variety of variations, but for the purpose of this study, we used the ViT-B32 model in each institution. For each ViT model, we added a flatten layer, a batch normalization layer followed by a dense layer of size 11, and then a layer of batch normalization followed by Softmax function to offer the probability of binary classification (DR or Normal). Non-FLViT was locally trained using the same model of FLViT hyperparameters and 100 epochs on the same data division.

## 3. Diabetic Retinopathy Datasets

In this section, we present the dataset used in each institution. The name of the institution has the same name as the dataset.

### 3.1. APTOS

The Asia Pacific TeleOphthalmology Society 2019 Blindness Detection (APTOS 2019) dataset's retinal images were used in this investigation. It is an open (Kaggle) competition called APTOS [42]. The related dataset includes 3662 retinal images that were gathered from various individuals who lived in India's rural areas. The RGB (Red–Green–Blue) images were taken with a Fundus camera. Following that, the samples were labelled by experienced medical professionals who divided the severity of blindness into five categories: no DR, mild DR, moderate DR, severe DR, and proliferative DR. The fundus images were taken in a variety of settings. Figure 2 shows some fundus images from the APTOS dataset.

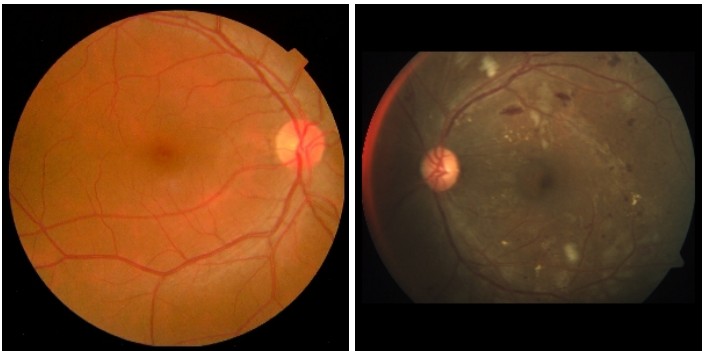

**Figure 2.** Example of fundus images from the APTOS dataset.

### 3.2. MESSIDOR-1 and MESSIDOR-2

A total of 1200 color images of the retinal fundus of the eyes were collected by three ophthalmological departments and are available in MESSIDOR-1 [39], and MESSIDOR-2 contains 1748 images. A Topcon TRC NW6 non-mydriatic retinograph equipped with a 3 CCD camera with a 45-degree field of view and resolutions of 1440 × 960, 2240 × 1488, and 2304 × 1536 was obtained. In this study, we merged the two datasets to obtain one set called MESSIDOR. Figure 3 shows some fundus images from the MESSIDOR dataset.

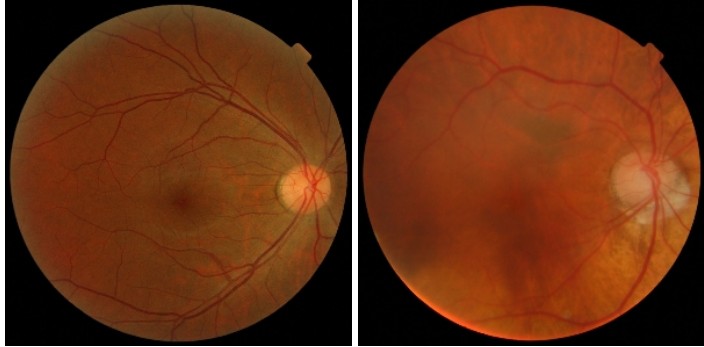

**Figure 3.** Example of fundus images from the MESSIDOR dataset.

### 3.3. IDRID

The dataset in [40] consists of 516 photos with a range of pathological DR circumstances; the images were all centred close to the macula and were taken with a Kowa VX-10 alpha digital fundus camera with a 50-degree field of view (FOV). The resolution of the images is 4288 × 2848 pixels. Each image was given a diagnosis by medical professionals, who also graded each image's retinopathy from zero (normal) to four (severe) and

determined whether or not DR was present. Figure 4 shows some fundus images of the IDRID dataset.

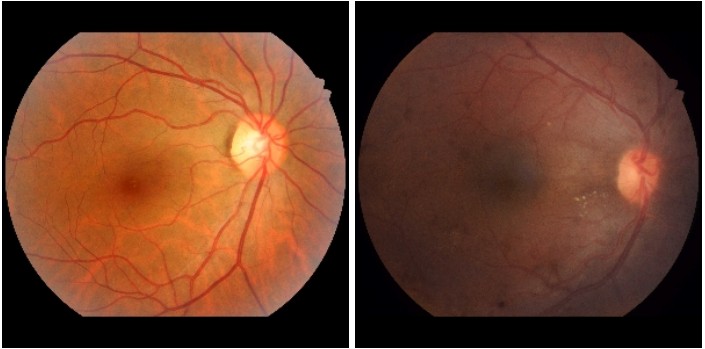

**Figure 4.** Example of fundus images from the IDRID dataset.

*3.4. Eyepacs*

The dataset in [41] contains about 88,702 high-resolution images taken under various imaging circumstances. These retinal images were taken from a set of individuals, and for each person, left and right eyes received two images apiece. The images were captured using various camera sizes and types, which may alter how left and right seem to the eye. This dataset is inconsistent since "proliferative DR" images make up a small percentage of the dataset whereas normal images with the label "0" represent an enormous class. An example of a fundus image from the Eyepacs dataset can be seen in Figure 5. Because the number of this base is greater than that of the other datasets, we took a sample of 6000 images to train and test the ViT models.

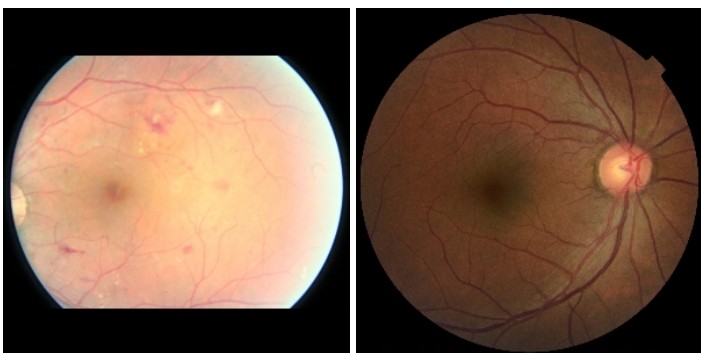

**Figure 5.** Example of fundus images from the Eyepacs dataset.

Table 1 offers a general summary of the dataset used in our study.

**Table 1.** Public datasets used for training and testing in our study.

| Name | Nbr. of Images | Resolution | Uses |
|:---:|:---:|:---|:---|
| **EyePACS** | 6000 | 1440 × 960<br>2240 × 1488<br>2304 × 1536<br>4288 × 2848 | DR grading<br>Exudates, Hemorrhage<br>and Microaneurysms detection |
| **MESSIDOR** | 1200 | 1440 × 960<br>2240 × 1488<br>2304 × 1536 | Exudates, Hemorrhage,<br>Microaneurysms and<br>abnormal blood vessel detection |
| **MESSIDOR-2** | 1748 | 1440 × 960<br>2240 × 1488<br>2304 × 1536 | Exudates, Hemorrhage,<br>Microaneurysms and<br>abnormal blood vessel detection |

**Table 1.** *Cont.*

| Name | Nbr. of Images | Resolution | Uses |
|:---:|:---:|:---:|:---:|
| **IDRID** | 516 | $4288 \times 2848$ | Exudates, Hemorrhage, Microaneurysms and abnormal blood vessel detection |
| **APTOS** | 3662 | $2124 \times 2056$ | Exudates, Hemorrhage, Microaneurysms and abnormal blood vessel detection |

## 4. Federated Learning

Deep learning (DL) models are frequently trained centrally, with client-site data kept, and model owners have access to the client data. Data sensitivity makes it difficult in many situations to gather diverse and comprehensive datasets. This makes it challenging to build strong deep learning models, which necessitate suitably vast and diverse datasets. FL, which decentralises the training of machine learning models, is presented by McMahan et al. [31] as a solution to this issue. Clients can participate in DL model training in FL by each training a model with a local dataset and then sharing the model parameters with other clients. The authors employ a method known as "Fedavg," which is a weighted average of the models. In our study, we use the Fedavg approach with ViT models for detecting DR. The global model is initially initialized in Fedavg. The current global model $wt$ is then sent by the central server to a chosen subset $C$ of all institutions $K$ in each round $t$. The set $S_t$ represents the chosen institutions. Each institution $k$ then updates its local model parameters and sends them to the server after training the model on its own local data $Pk$ to produce a model $w_t + 1k$. The server then uses the equation below to combine the weights of the received models to create a new global model,

$$w_{t+1} = \sum_{k \epsilon S_t} \frac{n_k}{n_t} w_{t+1}^k,$$

where $n_k$ is the number of samples at institution $k$ and $n_t$ is the total number of samples from all institutions. The server transmits the aggregated model to the institutions in the network after the training is completed.

## 5. Metrics

For performance evaluations, we used the following metrics: Accuracy (*ACC*), Sensitivity (*SN*), and Specificity (*SP*). These measures are described as follows:

$$SN = \frac{TP}{TP + FN}, \tag{1}$$

$$SP = \frac{TN}{TN + FP}, \tag{2}$$

$$ACC = \frac{TP + TN}{TP + FN + TN + FP}, \tag{3}$$

where $TP$ stands for the true positive rate and reflects the number of correctly labeled positive cases; $TN$ stands for the true negative rate and indicates the number of correctly labeled negative cases. The number of positive cases that are incorrectly labeled as positive is indicated by the $FP$ and the $FN$ indicators, respectively. Area under the curve (AUC), a performance statistic widely used for medical classification issues to show where the model compromises between accurate and incorrect diagnoses, is derived using the ROC curve.

## 6. Results

In this section, we report the DR detection using FLViT and Non-FLViT results from each institution.

Keras Library [49] was used to develop the FLViT. The training was carried out using Nvidia P6000 [50]. RectifiedAdam [51] was used as an optimizer, and the Batch size was fixed to 32 for each model with 100 rounds. All CXR images were resized to $224 \times 224$.

For APTOS institution, after 100 rounds using the FLViT technique, the model achieved a high ACC score of 0.95 and 0.94 for Non-FLViT which is a modest improvement by 1%. For MESSIDOR institution and after 100 rounds, the model obtained an ACC of 0.79 using the FLViT method and 0.76 for Non-FLViT (an improvement of 3%). This shows the model learned more important features by using the FLViT technique. Same in IDRID institution, the ACC score was improved by 2% when using the FLViT technique. The model provided an ACC score of 0.71 and 0.69 for Non-FLViT. The sensitivity score for detecting DR is higher when the model is trained on the FLViT technique. The model achieved a SN score of 0.85 vs. 0.77 for Non-FLViT. Eyepacs institution achieved an improved ACC score of 0.71 with FLViT and 0.68 for Non-FLViT. Unlike IDRID, the FLViT technique helps the model to improve the specificity with a score of 0.83 vs. 0.63 for Non-FLViT.

The confusion matrix for APTOS institution using FLViT is shown in Figure 6a, and Figure 6e presents the confusion matrix for Non-FLViT. As we can see, the model trained with the FLViT technique detects more true positives compared to Non-FLViT. The result is the same for MESSIDOR institution. The detection of DR is improved when using the FLViT technique, as we can see in Figure 6b. The FL model detects 112 cases for DR and 71 for Non-FLViT (see Figure 6f). The confusion matrix for IDRID institution using FLViT is shown in Figure 6c, and Figure 6g presents the confusion matrix for Non-FLViT. The model trained with the FLViT technique detects more true positives and true negatives compared to Non-FLViT. This shows that the FLViT technique helps the model to improve its performance. Figure 6d is the confusion matrix for Eyepacs institution using the FLViT technique. The model detects more DR cases compared to the Non-FLViT presented in Figure 6h.

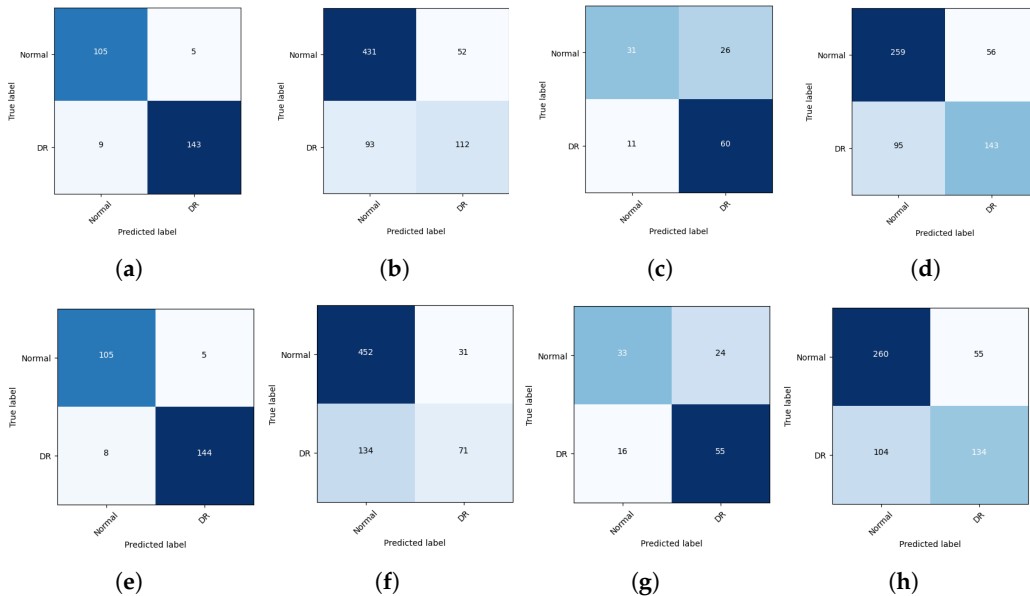

**Figure 6.** Confusion matrices of APTOS FLViT (**a**), Non-FLViT (**e**); MESSIDOR FLViT (**b**), Non-FLViT (**f**); IDRID FLViT (**c**), Non-FLViT (**g**); Eyepacs FLViT (**d**), Non-FLViT (**h**) (DR vs. normal) classification.

Figure 7 shows the AUC curves for APTOS institution, an AUC score of 0.95 for the FLViT technique and 0.94 for Non-FLViT (less than FLViT by 1%). For MESSIDOR institution (Figure 8), the model obtained an AUC score of 0.83 for FLViT and 0.74 for Non-FLViT. The result is the same for IDRID institution (see Figure 9). The FLViT model achieved an AUC score of 0.74

and 0.73 for Non-FLViT. On the Eyepacs institution, the model obtained an AUC of 0.77 when using the FLViT model and 0.76 for Non-FLViT (see Figure 10). This shows that the suggested technique offers a significant improvement compared to the trained local models.

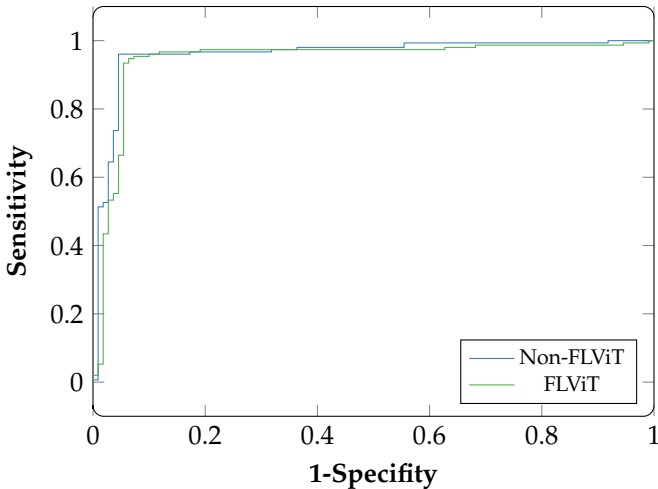

**Figure 7.** ROC curves of Non-FLViT and FLViT for APTOS institution classification (DR vs. normal).

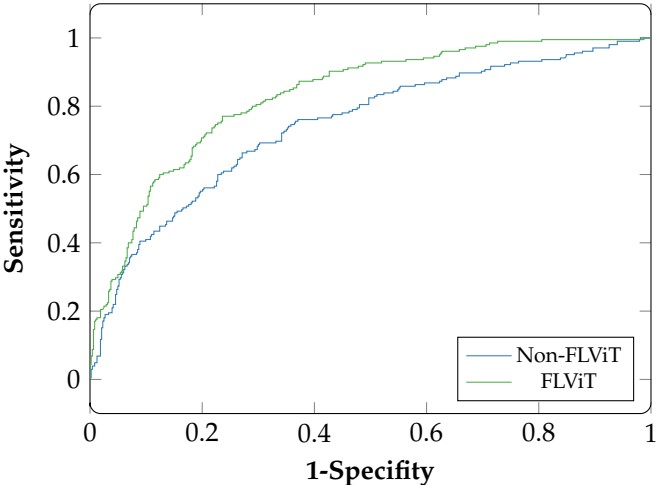

**Figure 8.** ROC curves of Non-FLViT and FLViT for MESSIDOR institution classification (DR vs. normal).

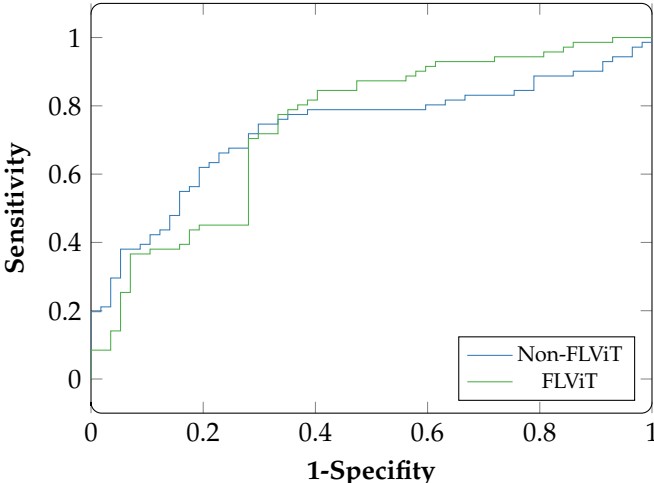

**Figure 9.** ROC curves of Non-FLViT and FLViT for IDRID institution classification (DR vs. normal).

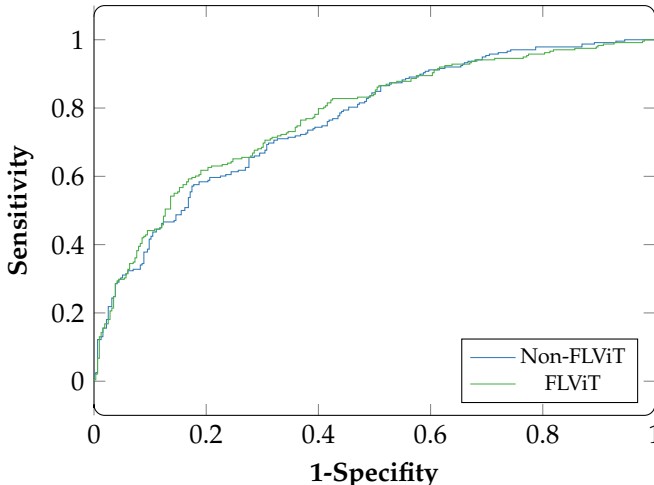

**Figure 10.** ROC curves of Non-FLViT and FLViT for Eyepacs institution classification (DR vs. normal).

Table 2 summarizes the performance scores obtained for each institution for FLViT and Non-FLViT techniques.

**Table 2.** Performance measures using FLViT vs. Non-FLViT with Vision Transformer models.

| | FLViT | | | | Non-FLViT | | | |
|---|---|---|---|---|---|---|---|---|
| Institutions | ACC | AUC | SP | SN | ACC | AUC | SP | SN |
| APTOS | 0.95 | 0.95 | 0.95 | 0.95 | 0.94 | 0.94 | 0.95 | 0.94 |
| MESSIDOR | 0.79 | 0.83 | 0.89 | 0.55 | 0.76 | 0.74 | 0.94 | 0.35 |
| IDRID | 0.71 | 0.74 | 0.54 | 0.85 | 0.69 | 0.73 | 0.58 | 0.77 |
| Eyepacs | 0.71 | 0.77 | 0.83 | 0.56 | 0.68 | 0.76 | 0.63 | 0.72 |

For a comparison with the CNN model, we chose one of the robust CNN models named DenseNet-121. The model offers interesting results in the classification of medical images according to several studies. Table 3 shows the results of this model with a federated (FL-CNN) and a non-federated learning (Non-FLCNN) technique. As we can see, the model obtains lower results compared to the transformer in terms of accuracy and AUC, and the sensitivity and specificity are almost close. This shows that the approach with Vision Transformers performs better than the CNN.

**Table 3.** Performance measures using FL-CNN and Non-FLCNN.

| | FL-CNN | | | | Non-FLCNN | | | |
|---|---|---|---|---|---|---|---|---|
| Institutions | ACC | AUC | SP | SN | ACC | AUC | SP | SN |
| APTOS | 0.94 | 0.93 | 0.93 | 0.93 | 0.91 | 0.91 | 0.92 | 0.94 |
| MESSIDOR | 0.80 | 0.83 | 0.90 | 0.53 | 0.74 | 0.73 | 0.93 | 0.37 |
| IDRID | 0.71 | 0.73 | 0.60 | 0.80 | 0.68 | 0.72 | 0.56 | 0.80 |
| Eyepacs | 0.67 | 0.72 | 0.87 | 0.51 | 0.65 | 0.76 | 0.66 | 0.68 |

## 7. Discussion

The demand for labeled ground-truth data considerably rises as deep learning applications become more complicated. A single client frequently lacks the resources to obtain the data required to build an accurate model. Additionally, medical images are securely protected by several privacy regulations, creating a considerable barrier to client's collaboration on data sharing. Additionally, it is possible that models developed primarily

using a single data island are greatly overfitted, which would restrict their ability to be applied to new data in the future. Federated learning offers a way to jointly train a model while maintaining the privacy of the image data.

In our study, we investigated the use of federated learning to expand the useful dataset size that frequently comes along with highly specialized studies in novel methodologies, such as disease classification on fundus images. Federated learning could make it easier for groups researching rare diseases to work together as there are fewer open-sourced datasets that are available to the public and more images that are kept inside of individual institutions. We concentrated on the use-case of federated learning that promotes multi-institutional collaborative studies towards more specialized research areas, even though there is merit for using it to increase the generalizability of tasks with widely accessible datasets like fundus imaging.

The performance of federated learning was better than that of internal models. When evaluated on data from the other domain and when a model was produced that outperformed those trained and tested at various sites, the findings for the classification problems indicated that all participants gained from federated learning. However, on the Eyepacs dataset, which contains more images than other datasets, the federated learning models underperformed the internal models. Further research into the impact of data distribution and uneven datasets on federated learning is warranted in light of this.

A model that can generalize to numerous datasets and perform as well as internal models was trained with the help of the federated learning framework. We predict that the federated models will perform better on fundus images from an unidentified source since they were trained on a more varied pool of data. The performance of each model on data obtained by devices produced by the same company is one area that needs more investigation.

The volume and variety of training data is a restriction of federated learning, much like for traditional deep learning. The Eyepacs dataset had a lot more images than the other three datasets, therefore one client trained by iterating over more steps per epoch. However, each client model was given identical weighting when averaging the aggregated client models. In spite of the data imbalance, this was performed to ensure that the federated model was not biased against one particular data source. The smaller datasets could be further enhanced as an option, but the benefits may be minor because each client model is combined into a global model.

Federated learning is employed to train distributed models across a variety of devices, including smartphones, wearable medical devices, automobiles, and Internet of Things (IoT) devices. They aid in the development of a strong model, but the training data are retained locally rather than being shared, resolving issues with data security, privacy, and access rights.

In the FL system, the learning process is typically orchestrated by a central server, which also updates the model based on client training outcomes. The fault tolerance of such a star-shaped server-client architecture is reduced: it does not address the issue of information governance, and it necessitates a powerful central server, which may not always be available in many real-world scenarios with a sizable number of clients [52,53].

Clients may be exposed to risks because of a server being present in the network. To compromise the entire local training group, the attacker could, for instance, utilize a fine-tuning approach to produce some malicious updates that are subsequently sent from the central parameter server. Federated learning systems that depend on a solitary parameter server also run a risk of failing. Training may be stopped or interrupted if the main parameter server is compromised by the attacker.

Thus, the completely decentralized FL was offered as a solution to the aforementioned issues, replacing client-to-client communication with peer-to-peer communication between linked clients [54].

## 8. Conclusions

In this work, we created a framework that enables numerous users to undertake federated learning on decentralized data. With the help of our findings, we were able to demonstrate that federated learning models can outperform the internal models, offering an effective means for increasing the data pool while protecting patient privacy. The suggested approach develops an accurate and cooperative DL based on the ViT model for multi-institution collaborations without risking data privacy, which is crucial in ophthalmology healthcare, particularly in ocular image processing. This was performed to learn ways to use FL successfully and efficiently in actual clinical situations. To obtain better outcomes, future work may include more clients, which would enhance the suggested technique even more. Additionally, we anticipate conducting comprehensive empirical investigations using different methodologies such as split federated learning to increase the degree of data privacy.

**Author Contributions:** Conceptualization, M.C. and M.A.A.; methodology, M.C. and M.A.A.; software, M.C.; validation, M.C. and M.A.A.; formal analysis, M.C. and M.A.A.; investigation, M.A.A. and M.C.; resources, M.C.; data curation, M.C.; writing—original draft preparation, M.C. and M.A.A.; writing—review and editing, M.C. and M.A.A.; supervision, M.A.A.; project administration, M.A.A.; funding acquisition, M.A.A. All authors have read and agreed to the published version of the manuscript.

**Funding:** This work was partially supported by the Natural Sciences and Engineering Research Council of Canada (NSERC), Alliance Grants (ALLRP 552039-20), New Brunswick Innovation Foundation (NBIF) COVID-19 Research Fund (COV2020-042), and the Atlantic Canada Opportunities Agency (ACOA), Regional Economic Growth through Innovation-Business Scale-Up and Productivity (project 217148).

**Institutional Review Board Statement:** IRB waived the approval requirements since the data used in this work were anonymized and public.

**Informed Consent Statement:** Not applicable.

**Data Availability Statement:** The data used in this work come mainly from public datasets, see Section 3 for more details.

**Conflicts of Interest:** The authors declare no conflict of interest.

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
