# Peer review of "Federated Learning for Diabetic Retinopathy Detection Using Vision Transformers"

_biomedinformatics, doi:10.3390/biomedinformatics3040058_

Round 1

Reviewer 1 Report

Comments and Suggestions for Authors

Dear Author/s Regarding the manuscript titled "Federated learning for diabetic retinopathy detection using Vision Transformers"

who presented a solution based on deep learning for the manual detection of retinal fundus images, which requires time, effort and cost and is prone to errors compared to computer-aided detection methods. The methodology used in this study is interesting and uses one of the new methods and cutting-edge technology.

The study is well described, the content of the manuscript is well organized. The manuscript has a strong backbone and deserves to be published in this journal with a few edits. We suggest that you carefully pay attention to the following points so that it can be published.

1: The abstract needs substantial editing. Mention all findings carefully in the abstract. Mention the performance evaluation metrics in the abstract.

2: In the abstract, the conclusion part is faint. Add one or two lines of conclusion at the end of the abstract to close the abstract.

3: It is suggested to have a section for a brief reference to the deep learning method under the introduction section. We suggest you use the following articles and give them a website for the richness of the bibliography of the references.

"Deep convolutional neural network–based computer-aided detection system for COVID-19 using multiple lung scans: design and implementation study"

"Management of covid-19 detection using artificial intelligence in 2020 pandemic"

"Deep learning: Applications, architectures, models, tools, and frameworks: A comprehensive survey"

"A mobile application based on efficient lightweight CNN model for classification of B-ALL cancer from non-cancerous cells: a design and implementation study"

4: It is suggested to write brief suggestions for future studies in the conclusion section. This issue will be a bright light for future studies.

Author Response

Thank you for taking the time to review our article, you will find attached the response of your important comments.

Reviewer 2 Report

Comments and Suggestions for Authors

Chetoui and Akhloufi proposed a federated learning approach to classify diabetic retinopathy based on vision transformers. The authors claimed no studies have ever integrated the transformer vision models into a federal learning framework to detect diabetic retinopathy. The authors show an improvement of up to 3% accuracy with the proposed federated learning strategy. However, I have several comments that need to be addressed by the authors.

Major:

1.      The authors do not present much detail as to how the federal learning was tuned, other than to say. For example, how the training set, testing test, and validation set are split on each local institute? How the FLViT and Non-FLViT were compared? Are there a “global validation set” for all models to compare?

2.      Many details about how the proposed federal learning approach would be used and the limitations of FLViT are not covered in sufficient detail in the paper, omitting compelling evidence needed for deployment.

3.      The gain in accuracy compared to a model locally trained on several institutions is not very significant (up to 3% accuracy, Table 1), which would make it unlikely that the FLViT gains from participating in the federal learning protocol.

4.      The authors chose the ViT model as the architecture because it’s “SOTA”  However, it is still necessary to have a CNN-based mode (VGG, ALEXNET, RESNET etc ) to show the superior performance of FLViT vs FL CNN.  

Minor:

1.      The experimental design for the data distribution is definitely good. In the field of federated learning, training on data with non-IID distribution has long been a challenge. In the current experiment design, the non-IID distribution comes from factors such as data generation technologies. I would suggest, if the data are available, include an experiment that divide data into different institute according to age, gender or other demographic features will help further prove the robustness of the method.

2.      Section 3 of “Diabetic Retinopathy Datasets” is mainly about the data resource. There are 4 independent datasets mentioned here. I suggested making a main Table 1 to list the essential information, make sure the table is clear, and will be a good benefit for understanding the downstream analysis in this study.:

Author Response

(The authors gave the same response as above.)

Round 2

Reviewer 2 Report

Comments and Suggestions for Authors

No further comments